# PeerJ

# PANDA: pathway and annotation explorer for visualizing and interpreting gene-centric data

Steven N. Hart[1,3], Raymond M. Moore[1,3], Michael T. Zimmermann[1], Gavin R. Oliver[1], Jan B. Egan[2], Alan H. Bryce[2] and Jean-Pierre A. Kocher[1]

[1] Division of Biomedical Statistics and Informatics, Department of Health Sciences Research, Mayo Clinic, Rochester, MN, USA
[2] Division of Hematology/Oncology Mayo Clinic, Mayo Clinic Cancer Center, Scottsdale, AZ, USA
[3] These authors contributed equally to this work.

## ABSTRACT

**Objective.** Bringing together genomics, transcriptomics, proteomics, and other -omics technologies is an important step towards developing highly personalized medicine. However, instrumentation has advances far beyond expectations and now we are able to generate data faster than it can be interpreted.

**Materials and Methods.** We have developed PANDA (Pathway AND Annotation) Explorer, a visualization tool that integrates gene-level annotation in the context of biological pathways to help interpret complex data from disparate sources. PANDA is a web-based application that displays data in the context of well-studied pathways like KEGG, BioCarta, and PharmGKB. PANDA represents data/annotations as icons in the graph while maintaining the other data elements (i.e., other columns for the table of annotations). Custom pathways from underrepresented diseases can be imported when existing data sources are inadequate. PANDA also allows sharing annotations among collaborators.

**Results.** In our first use case, we show how easy it is to view supplemental data from a manuscript in the context of a user's own data. Another use-case is provided describing how PANDA was leveraged to design a treatment strategy from the somatic variants found in the tumor of a patient with metastatic sarcomatoid renal cell carcinoma.

**Conclusion.** PANDA facilitates the interpretation of gene-centric annotations by visually integrating this information with context of biological pathways. The application can be downloaded or used directly from our website: http://bioinformaticstools.mayo.edu/research/panda-viewer/.

Corresponding author
Jean-Pierre A. Kocher,
kocher.jeanpierre@mayo.edu

## BACKGROUND AND SIGNIFICANCE

The development of high throughput technologies is a major driver in the development of personalized medicine. The ability to rapidly and accurately interrogate individuals' disease states at the molecular level has revealed a diversity of personal gene alteration

landscapes and expression profiles between and within pathologic conditions (*Weinstein et al., 2013*). This diversity translates to markedly differing disease characteristics, creating the requirement for individually tailored treatment strategies to reach optimal therapeutic effect. However, a gap exists in our ability to translate identified alterations into information that can be interpreted by clinical researchers. This translation requires prioritizing alterations based on disease and clinical relevance. While it is conceptually straight forward to limit analysis to the genes for which a clinical course could be taken, oftentimes the biology is more complex. Some driver mutations are best targeted by drugs which affect genes downstream of the driver itself. For example, a large proportion of clear cell renal cell carcinoma (ccRCC) is driven by loss of VHL (*Foster et al., 1994*; *Shuin et al., 1994*), a gene which is not directly druggable. However, all FDA approved drugs for ccRCC target downstream genes in the VHL pathway, either through VEGF or mTOR pathways (*Molina, Motzer & Heng, 2013*). A similar scenario is from GNAQ or GNA11 mutant melanomas, where treatment with MEK inhibitors has demonstrated efficacy (*Carvajal et al., 2013*). Thus, it is imperative that variants be considered in context of the affected pathways, and not just as isolated phenomena.

So the question becomes, how can one view data in the context of pathways? There are several tools that exist to explain data in the context of biological pathways, including, but not limited to Cytoscape (*Cline et al., 2007*), DAVID (*Huang, Sherman & Lempicki, 2009*), and WebGestalt (*Wang et al., 2013*). DAVID and WebGestalt are both web-based applications that can be used to upload gene lists and test for significant enrichment—displaying the outputs in the form of tables. DAVID can go one step further if the resulting gene set reaches statistical enrichment in that it can link out to a KEGG pathway with flashing icons next to the genes in the list. The benefit of this approach is that the genes are highlighted while maintaining the functional topology (i.e., their biological order in the reactions) which is helpful in understanding downstream biological effects. The downsides are two-fold. First, there is no context to the data—just a gene list. If the gene list was describing the results of a gene expression study, then the expression level, probe id, or any other relevant information would not be persisted for the user. Second, users are limited to viewing one gene list at a time. If they were to combine gene lists (say for example genes with mutations and genes with altered expression), then there would not be a way to discriminate between which list the gene originally came from. The other visualization tool is Cytoscape, which is a downloadable program designed to work on networks of genes. A user could upload a list of genes which the program displays as a set of nodes in a graph. Continuing with the example of users with gene expression as before, users can change the node shape or color to identify the gene as being mutated or overexpressed. The coloration or shape of each node only represents a binary representation (i.e., was the gene mutated/overexpressed or not), so any associated information like what is the type of mutation or degree of overexpression is not available. The limited number of display features one can manipulate in Cytoscape to describe events quickly become evident when users also want to see down-regulated genes, genes that are druggable, genes that are disease associated, etc. While Cytoscape is a powerful tool for bioinformaticians, there is a

steep learning curve to become useful for new users. Also, nodes are no longer represented in the topology of their biological pathways, but rather in what is commonly referred to as a "hairball," making it difficult the downstream biological impact of the effect they are observing.

To address this issue, we have developed a software solution called PANDA (Pathway AND Annotation) Explorer. PANDA enables the visualization of genomics and drug information in the context of pathways. It is a support tool designed to help clinical researchers integrate data (e.g., genomics alterations) and annotations (e.g., available drug treatments) to strategize therapeutic treatments for individual patients or to understand the disease biology. PANDA differs from other pathway visualization tools in many ways. First, PANDA is a simple to use web application with an intuitive graphical user interface. Second, PANDA is capable of combining annotation sets (genomics and drug information) and pathway informatics within the same display while minimizing clutter in the visual field. Third, PANDA includes an authentication and data sharing mechanism to facilitate collaborations between clinicians, scientists, bioinformaticians, or their support teams (such as a Tumor Board). Finally, PANDA can perform pathway-level enrichment analysis. PANDA is available at http://bioinformaticstools.mayo.edu/research/panda-viewer/.

## MATERIALS AND METHODS

### What is PANDA?

We have developed a genomics results reporting tool called PANDA (Pathway AND Annotation) explorer. PANDA enables the visualization of multiple annotations in the context of pathways. Annotations in this context are a broad term that refers to various genomics features and information. Annotations can be one of three modalities. First, they can be extracted from a biospecimens such as SNVs, CNVs, structural variants, or gene expression. Second, annotations can be information extracted from public or internal data sources (*Abecasis et al., 2012*; *Sherry et al., 2001*) such as frequency of variants, known associations between gene, diseases, and drug-gene relationship. Finally, annotations could also be predictions by software applications (reviewed in *Wu & Jiang, 2013*) such as the deleterious nature of a mutation. The number of possible annotations is innumerable, and each type may require its own details to make it useful. This diversity makes it very difficult to model all annotations under a traditional method. For instance, genomic information of clinical relevance can include the number of variants in the gene, the position and frequency of these variants in the general population, and the nature of the variant (deleterious or benign). However, these variant-centric annotations can be difficult to reconcile with gene-centric annotations such as the expression level of a gene, the methylation status of a gene profile of a gene, the druggability of a gene, etc. Instead, PANDA summarizes annotations at the gene level and uses an innovative icon-based representation to display these features on biological pathways maps. The use of icons reduces the cluttering of the display, facilitating the visual integration and interpretation of annotations with pathways information. For instance, the relationship between a mutated oncogene that results in a downstream gene becoming up regulated can easily be spotted.

PANDA is also designed to help bioinformaticians deliver gene-centric results in a form more readily interpretable by researchers and clinicians. The application assumes that the inputs have undergone quality and disease-relevance filtering so that it only displays relevant information. PANDA includes an authentication and access-control mechanism to facilitate sharing of dataset between team members and collaborators. Altogether, the tool allows users to select and visualize pathways, toggle annotations views, perform enrichment analysis, and authorize sharing of data with collaborators.

PANDA differs from other pathway visualization tools because users can upload and visualize any number of annotations with any type of content. Icons can be selected to represent a type of annotation (e.g., mutation, expression, etc.) in order to provide visual cues as to what the data represents which is helpful when there are multiple annotations loaded. PANDA also links genes to GeneCards and pathway-level enrichment analysis can be performed on the fly. For convenience, PANDA is pre-loaded with several commonly used annotation sources. This includes 19,777 gene-drug relationships from the Drug Gene Interaction Database (DGIDB) (*Griffith et al., 2013*), 5,002 gene entries from MalaCards (*Rappaport et al., 2013*), 3,945 genes from Online Mendelian Inheritance in Man (OMIM, http://omim.org/), 3,243 genes from Human Phenotype Ontology (HPO) (*Kohler et al., 2014*), and 56 genes from the Pharmacogenomics Knowledgebase (PharmGKB) (*Whirl-Carrillo et al., 2012*). Details and the code used to generate these annotations are available on our GitHub site.

## Input file format for annotation and annotation sets

Since PANDA is not exclusive to any single technology platform (e.g., proteomics, gene expression, DNA sequencing, etc.), there are an immense number of ways that data and annotations can be represented. The data sources are often large and highly complex, thus requiring bioinformaticians to preprocess, annotate, and filter data using appropriate methods for the study. As such, we have designed PANDA to accept as input a simple tab-delimited file format. Each file requires a gene symbol in the first column, followed by one or more annotation field(s) that will be displayed to the user in a later step. Adding a "#" sign to the header line ensures that they table header is transferred to the pathway level view. Each uploaded file is one source of annotation and is assigned a single icon to represent the underlying data.

## Operation

### *Login*

Figure 1 and the following text describe how to navigate through the application. PANDA includes an authentication mechanism equipped with logging in and verifying passwords. Users are required to register (which is free) before logging into PANDA. This authentication mechanism is coupled to the access control and data sharing feature (see below), so a user must be registered in the system before data can be shared with them.
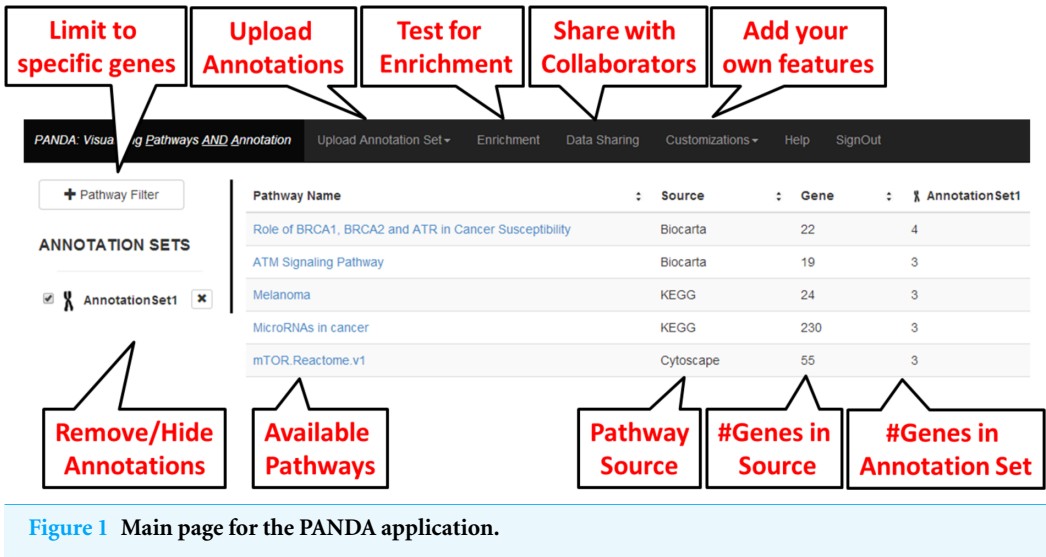

**Figure 1** Main page for the PANDA application.

## Main page

PANDA's main page is displayed as a table of 342 BioCarta 168 KEGG (*Kanehisa et al., 2008*; *Kanehisa & Goto, 2000*; *Kanehisa et al., 2012*), and 92 PharmGKB preloaded pathways, along with the total number of genes in the pathway. For each annotation uploaded by the user, additional columns are appended to the right of the table, displaying the total number of genes in each pathway that are included in the annotation. On the upper left of the main window, a Pathway Filter feature lets users restrict the number of pathways displayed to the ones including genes of interest. For example, if a user wanted to know which pathways the *KRAS* gene was in, they could restrict the table to just contain those pathways. A set of navigation tabs is displayed at the upper level of the main page. Details on these different tabs are provided below.

## Uploading Annotations

Annotations are uploaded via the 'Upload Annotation Set' navigation tab. When uploading an annotation file, the first 10 lines are previewed to allow the user to validate that the first column is the gene symbol. In the second step of the upload process, the user assigns an icon to represent the annotation. Only one icon can be assigned to the annotation included in a file, but the same icon can be assigned to multiple annotations. It is recommended for continuity that icons be used consistently to represent identical data types whenever possible. For example, mutation events should always use the same icon, gene expression should be the same icon, and drugs should be the same icon, etc. The rationale for this is that when viewing the pathway, it becomes visually apparent what type of data exists within the pathway, without needing more detail.

Another useful feature is that users are allowed to upload their own icons to represent their data. In this way, they can assign a different icon to the dataset that has intrinsic meaning to them—more so than the prepopulated icons provided in the application itself. The option to upload user-specific icons is located under the 'Customize' tab.

During the upload process, some genes in the annotation file may not correspond to any of the genes that are listed in pathways. These genes are presented to the user after loading completes via a downloadable text file. Incorrect gene symbols are also recorded in this file.

## Annotation counts and enrichment analysis

Once the annotation files are uploaded, the main page displays the number of annotated genes per pathway per annotation. These columns, like any other column, can be sorted to quickly view the pathways with the largest or smallest number of annotated genes. To identify pathways that have more genes annotated than would be expected by chance, enrichment analysis can be performed on each uploaded annotation dataset, using the function located under the 'Enrichment' tab. The end result is an additional column in the main table containing the corresponding $p$-value from a Fisher's Exact test.

### Pathway viewer

Each pathways listed in the main page can be selected for visualization, regardless of whether or not it contains any uploaded annotations. The selected pathway is displayed in the 'Pathway Viewer' page. Icons representing each annotation set are display next to the associated gene. The annotation detail summarized by the icon is displayed by clicking or hovering the cursor over the icon. Clicking on any gene in the pathway will open the corresponding GeneCards webpage in new tab. The pathway viewer facilitates the visual integration of annotations in the topological context of interacting genes.

### Data sharing

To facilitate case review by peers or the clinician's team, a data sharing feature is provided via the 'Data Sharing' navigation tab. Annotation sets to share can be selected under this tab. In order to share data, the user must create a group, add members to that group, and select which annotations to share. Data can only be shared among registered users; the user must know the other user IDs to share the data.

### Custom pathways

The 'Custom' tab lets the user adjust or update some of PANDA's features. This is where PANDA allows the user to upload and remove their own images to be used as icons. Icons can be uploaded in the form of ".png," ".jpg," or ".gif." Similarly, custom pathways extracted from Cytoscape (*Cline et al., 2007*) can also be added to PANDA for annotation and visualization. This feature enables pathways to be included that are underrepresented in the existing sources. In this case two files are needed: a XGMML file that describe the pathway and ".png" file that provides an image of the pathway. Both files can be extracted from Cytoscape following the procedure described on our website.

### Hidden feature: gene normalization

Gene symbols listed in the first column of the annotation file are normalized during the uploading process in PANDA. Gene symbols are matched against the 'approved symbol' of HGNC a gene name reference database commonly used by pathways and other network applications such as Cytoscape If a gene symbol cannot be matched, a second phase of matching occurs against a list of HGNC 'synonyms'. If a match is found, the 'approved

symbol' is assigned to the gene. It should be noted that occasionally, a HGNC 'synonyms' can be associated to multiple HGNC 'approved symbols'. To avoid confusion, HGNC 'synonyms' are removed from the HGNC database stored in PANDA if they mapped to more than one HGNC approved symbol.

## RESULTS

### Use case 1: quickly comparing one's own data to a published set

Papers are commonly presenting large datasets as supplemental materials. An example is a paper we published previously in a study of pancreas cancer (*Murphy et al., 2013*). Supplemental table 2 of that paper shows the insertions and deletions per sample. Now let's say a user is interested in finding out if any of those mutated genes are known to OMIM, HPO terms, and subsets of their own data. Once the table is downloaded, users simple need to rearrange the "Gene" column to be the first, renaming the column header from "Gene" to "#Gene," choosing which other columns they would like to persist, and saving as a tab-delimited format. Once loaded, any genes in common between the user's dataset and from the supplemental material will now be represented with two icons next to those genes, instead of just one.

### Use case 2: presenting and sharing information in a clinical research setting

PANDA has proven valuable in the genomic oncology clinic at our institution. In the Individualized Medicine clinic, patients with advanced malignancies with limited standard treatment options can undergo next generation sequencing of their tumor in an attempt to find targetable variants. The level of sequencing can vary from limited gene panels of 50–200+ genes at one extreme, up to combined whole genome sequencing (WGS), RNA sequencing (RNA-Seq), and array CGH (aCGH) at the other. Once the sequencing is completed, the data is filtered through various bioinformatics pipelines and discussed at a Genomic Tumor Board (GTB). Only significant results from copy number assessment, differentially expressed genes, or relevant annotations are provided as input into PANDA so that the clinicians are not overwhelmed by trying to view all the raw data from different experiments simultaneously. The GTB then discusses the relevance of the various targets and attempts to create a treatment plan for the patient.

As an example, PANDA was used in evaluating the genome and transcriptome of a 55-year-old Caucasian male with metastatic sarcomatoid RCC with pulmonary metastases. Imaging demonstrated a large renal mass, retroperitoneal lymphadenopathy, and pulmonary masses. A biopsy of the kidney lesion established the histology. The patient elected to undergo genomic analysis of the tumor with WGS (tumor and germline), RNA-Seq, and aCGH. The aCGH showed amplification of YAP1, while WGS demonstrated P287T variant of CCND1 with evidence of possible allele specific expression by RNA-Seq. Figure 2 shows how the data are displayed for all of the assays performed on this patient. This combination of abnormalities was particularly intriguing as YAP1 amplification has been shown to drive CCND1 transcription (*Mizuno et al., 2012*) and the P287T

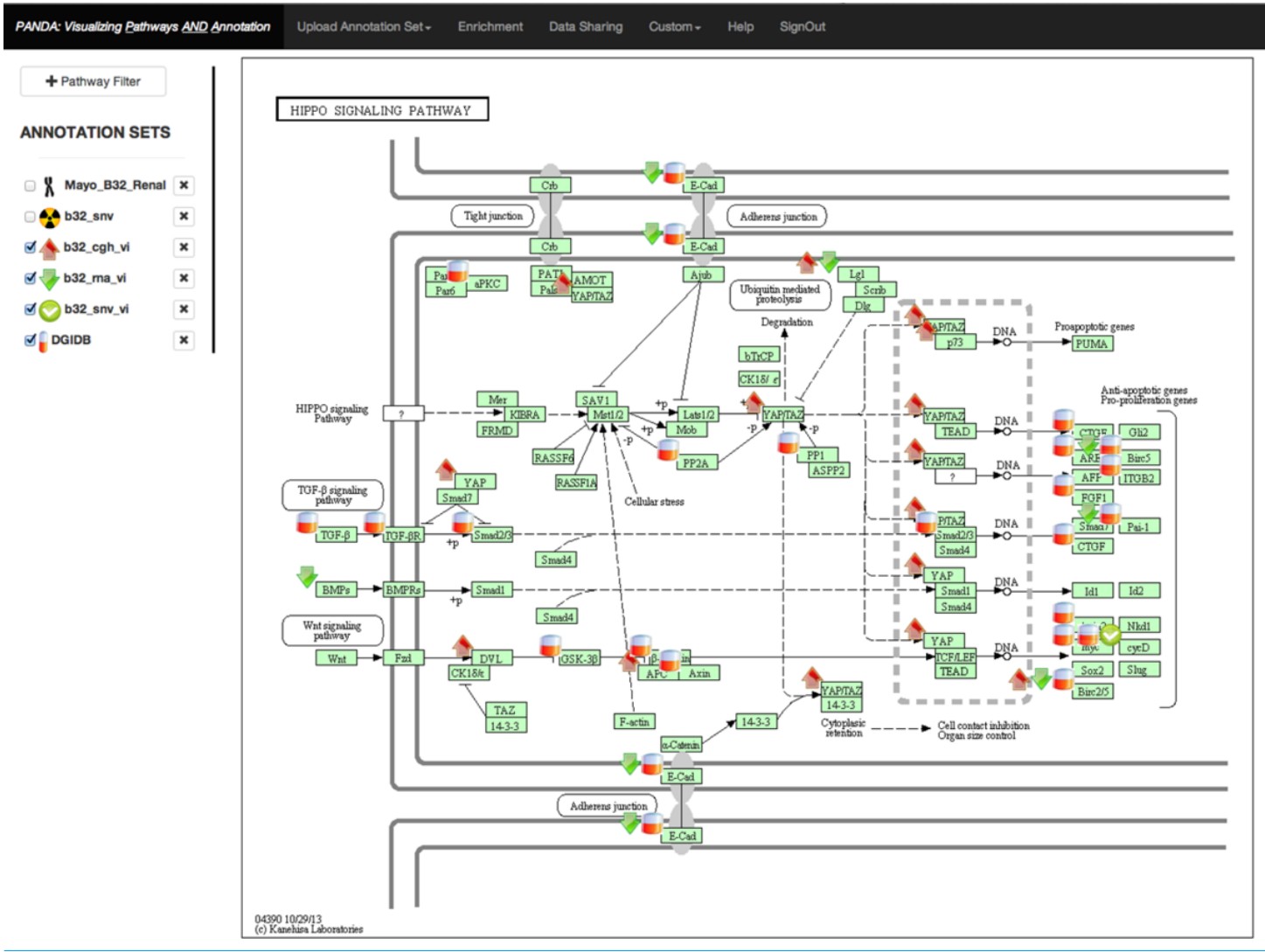

**Figure 2** **Example display of the Hippo Kinase pathway from KEGG.** Icons on the left and within the pathway represent different data types and annotations. The mouse cursor is hovering over the pill icon, which contains druggability information. On hover, the grey box appears showing the data contained within the "Drugs" file.

variant is hypothesized to inhibit polyubiquitination of CCND1, thereby inhibiting its degradation and promoting tumorigenesis (*Moreno-Bueno et al., 2003*). CCND1 activity is therapeutically targetable by inhibition of CDK4/6 (*Musgrove et al., 2011*), a target for which multiple agents are currently in clinical trials. The tumor also had multiple other potentially relevant variants including amplification of BIRC3, point mutations in ATM, and a splice variant of TP53. However, the presence of two variants both amplifying the same pathway formed the most compelling narrative for a driver pathway in this tumor, ultimately forming the basis for our treatment recommendation to start a CDK4/6 inhibitor.

## DISCUSSION

PANDA is designed to facilitate the interpretation of 'omics' data for individualized medicine and to provide a visual aid to clinical teams designing rational therapeutic treatment for individual patients. PANDA is intentionally designed to be simple to use by non-bioinformatics experts through an easy-to-use web interface and simple text file loading. The limited number of features reduces the number of pages to navigate, thereby decreasing the learning curve, making interpretation of the data the focus of the application. The use of icons to summarize these annotations significantly simplifies the visual field thereby enhancing interpretation of data within the context of biological pathways. This display approach can provide a fast overview of the deregulated or mutated genes and the drugs that target these genes or interacting genes. PANDA has proven useful in helping interpret the mutational landscape in patients and designing drug treatments.

Since PANDA uploads annotations in a tab delimited input format, no special software or complicated input files are required, and as such can easily be integrated into any data processing flow. The workflow implemented in our institution starts from the preprocessing of the genomics data, calling of variants, annotation of altered genes using BioR (*Kocher et al., 2014*) and prioritization of altered genes by a team of experts including bioinformaticians, biostatisticians, and genetic counselors. The final list of actionable altered genes and related annotation are then presented using PANDA to the clinicians on the Genomic Tumor Board to strategize drug treatment for a patient. The access control and data sharing mechanism implemented in PANDA facilitates collaboration among clinicians and other members of their scientific team. It also reduces the clinician's burden of uploading annotations assigning icons and managing the data since access can be provided to the support team that can easily handle these tasks.

In summary, PANDA is a tool that allows multiple pieces of data and information to be integrated into a more manageable graphical representation. Maintaining network topology structure makes understanding the up and downstream implications easier to digest. Our use of icons to represent large blocks of data types greatly simplifies the visual field, while still making the details available on-demand when they need to be viewed.

### Funding

We would like to acknowledge the Center for Individualized Medicine at Mayo Clinic for funding this research project. The funders had no role in study design, data collection and analysis, decision to publish, or preparation of the manuscript.

### Grant Disclosures

The following grant information was disclosed by the authors:
Center for Individualized Medicine.

### Competing Interests

The authors declare there are no competing interests.

## Author Contributions

- Steven N. Hart conceived and designed the experiments, performed the experiments, analyzed the data, wrote the paper, prepared figures and/or tables, reviewed drafts of the paper.
- Raymond M. Moore, Michael T. Zimmermann and Gavin R. Oliver performed the experiments, analyzed the data, wrote the paper, reviewed drafts of the paper.
- Jan B. Egan and Alan H. Bryce performed the experiments, analyzed the data, contributed reagents/materials/analysis tools, wrote the paper, reviewed drafts of the paper.
- Jean-Pierre A. Kocher conceived and designed the experiments, performed the experiments, analyzed the data, contributed reagents/materials/analysis tools, wrote the paper, reviewed drafts of the paper.

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
