# Peer review of "PANDA: pathway and annotation explorer for visualizing and interpreting gene-centric data"

_PeerJ, doi:10.7717/peerj.970_

## Round 0.1 · original submission · Minor Revisions

Please do address the reviewers concerns. In particular, do make sure to address questions regarding source code availability, licensing, etc. In addition, a response should be provided to Reviewer 1’s question of how to evaluate output from complex data sets involving multiple icons. Whether this is built into the software or just an outline of approaches used by the authors, guidance on how to analyze results would be beneficial to readers.

Reviewer 1 ·

Basic reporting

The article meet all PeerJ standards.

Experimental design

1. Although using different icons can integrate multiple omics data, it is difficult to find patterns in a pathway when the number of icons for each gene is over 3. Thus, the authors should provide some methods to help users find patterns in the pathway. For example, if users select mutation and gene expression features, PANDA should highlight some regions in a pathway that enrich with mutated genes or overexpressed genes or highlight several paths to show the upstream mutated genes affect downstream overexpressed genes. Because the advantage of PANDA is it can visualize omics data and pathway topology together, the traditional enrichment analysis is not useful for PANDA.
2. If the uploaded files contain multiple annotations, PANDA should assign different icons to different annotations. But, currently, all annotations can only be represented by one icon.
3. The authors should provide some examples in the website.

Validity of the findings

No validity problem.

Reviewer 2 ·

Basic reporting

The authors report on the creation of ‘PANDA’ an annotation and visualization tool the helps place ‘-omics’ data in the context of pathways, diseases, and drugs. The paper is well written and the authors do a good job of explaining the need for PANDA. They also do a good job of placing PANDA in the context of existing tools. I have seen applications that were conceptually similar to PANDA in a few places, but all of these were closed source, commercial products. If the authors release the code for this tools under an open source license and maintain a public instance that researchers can use for free, the impact of PANDA could be substantial.

Experimental design

PANDA seems to be well implemented and in my testing worked smoothly.

Validity of the findings

This is not an experimental paper so this review category seems forced. However, the authors do provide two practical use cases and do a nice job of illustrating how a user of PANDA would benefit from its results.

Additional comments

This is a great idea and a nice implementation of that idea.

Major issues:

In the abstract the authors describe displaying data in the context of pathways but do not provide examples of what kinds of data. The authors should list the data sets they currently support as the linking of disease and druggability information to pathways is a impressive and will increase the likelihood of a potential user actually reading the rest of the article and going to the web service.

Can the authors please deposit their source code in a version tracking system (e.g. github?) with an appropriate license? This is becoming industry practice for bioinformatics software and could enable the community to contribute to PANDA or fork it and build something on top of it.

Speaking of which, I do not see a license in the current source code tarball. The application is pretty slick and seems to be well implemented as a ruby/rails app. Do the authors plan on releasing this project open source? Until there is a license this aspect of the future plans for the project is ambiguous.

Perhaps I missed this, but can the authors comment on how they will update the preloaded annotation content in PANDA. Source like HPO, DGIdb, etc. are constantly being updated. Does the implementation use the APIs of these sources to update the content? Or are they static data dumps that would have to be painfully maintained for each case. A description of how data was obtained from each of these sources would be a good addition to the materials and methods section.

Minor issues:
The manuscript is for the most part well written. I found a few minor corrections detailed below.

On page 1 of the reviewer version of the manuscript, “However, instrumentation has advances far beyond …” should read “However, instrumentation has *advanced* far beyond …”

On page 2, “such as deleterious nature of a mutation.” should read “such as the deleterious nature of a mutation.”

On page 6, “… we have PANDA is pre-loaded with several commonly used annotation sources” should read: “PANDA is pre-loaded with several commonly used annotation sources”.

The authors should probably do one last check for further minor language issues like this.

---

## Round 0.2 · accepted · Accept

Thank you for addressing the reviewers' questions and comments.